# Defining the 3′Epigenetic Boundary of the *FMR1* Promoter and Its Loss in Individuals with Fragile X Syndrome

**DOI:** 10.3390/ijms241310712

**Published:** 2023-06-27

**Authors:** David E. Godler, Yoshimi Inaba, Minh Q. Bui, David Francis, Cindy Skinner, Charles E. Schwartz, David J. Amor

**Affiliations:** 1Diagnosis and Development, Murdoch Children’s Research Institute, Royal Children’s Hospital, Melbourne, VIC 3052, Australia; inachoku@hotmail.com; 2Department of Paediatrics, Faculty of Medicine, Dentistry and Health Sciences, University of Melbourne, Parkville, VIC 3052, Australia; david.amor@mcri.edu.au; 3Centre for Epidemiology and Biostatistics, Melbourne School of Population and Global Health, University of Melbourne, Melbourne, VIC 3052, Australia; mbui@unimelb.edu.au; 4Victorian Clinical Genetics Services and Murdoch Children’s Research Institute, The Royal Children’s Hospital, Melbourne, VIC 3052, Australia; david.francis@vcgs.org.au; 5Center for Molecular Studies, J.C. Self Research Institute of Human Genetics, Greenwood Genetic Center, Greenwood, SC 29646, USA; cskinner@ggc.org (C.S.); charles.schwartz224@gmail.com (C.E.S.); 6Neurodisability and Rehabilitation, Murdoch Children’s Research Institute, Royal Children’s Hospital, Melbourne, VIC 3052, Australia

**Keywords:** fragile X syndrome, FMR1, epigenetic boundary, premutation, methylation intellectual disability

## Abstract

This study characterizes the DNA methylation patterns specific to fragile X syndrome (FXS) with a full mutation (FM > 200 CGGs), premutation (PM 55–199 CGGs), and X inactivation in blood and brain tissues at the 3′ boundary of the *FMR1* promoter. Blood was analyzed from 95 controls and 462 individuals (32% males) with FM and PM alleles. Brain tissues (62% males) were analyzed from 12 controls and 4 with FXS. There was a significant increase in intron 1 methylation, extending to a newly defined 3′ epigenetic boundary in the FM compared with that in the control and PM groups (*p* < 0.0001), and this was consistent between the blood and brain tissues. A distinct intron 2 site showed a significant decrease in methylation for the FXS groups compared with the controls in both sexes (*p* < 0.01). In all female groups, most intron 1 (but not intron 2 sites) were sensitive to X inactivation. In all PM groups, methylation at the 3′ epigenetic boundary and the proximal sites was significantly decreased compared with that in the control and FM groups (*p* < 0.0001). In conclusion, abnormal *FMR1* intron 1 and 2 methylation that was sensitive to X inactivation in the blood and brain tissues provided a novel avenue for the detection of PM and FM alleles through DNA methylation analysis.

## 1. Introduction

Fragile X syndrome (FXS) is the most common heritable form of intellectual disability and is the second-most common cause of comorbid autism (1 in 3600 males and 1 in 6000 females) [1]. FXS usually results from expansions of a trinucleotide CGG repeat in the 5′ untranslated region (UTR) of the *FMR1* gene to ≥200 repeats, called full mutation (FM), with the normal range being <45 repeats [2]. The FM is associated with a decrease in the expression of *FMR1* due to promoter methylation, with subsequent loss of its protein product (FMRP), which is important for normal neurodevelopment [3,4]. FM males with mosaicism for CGG size and methylation have been reported to have better intellectual function than those without mosaicism [5]. In addition, approximately 5% of males with an FM have been reported to have a completely unmethylated promoter (UFM), associated with the expression of *FMR1* mRNA and FMRP, and an IQ within the normal range [6,7,8]. Smaller CGG expansions, defined as premutation (PM, 55–199 repeats), do not cause FXS but have been associated with elevated levels of *FMR1* mRNA linked to RNA toxicity [9]. These PM alleles are more common, with frequencies reported between 1 in 400 and 1 in 800 in males and between 1 in 300 and 1 in 400 in females [10]. The RNA toxicity associated with PM alleles has been linked to late onset disorders, fragile X-associated tremor/ataxia syndrome (FXTAS), and fragile X-associated primary ovarian insufficiency (FXPOI) [10,11,12]. However, the CGG size and *FMR1* mRNA levels have been shown to have limited utility as predictors of which individuals with PM alleles become symptomatic [11]. It has also been shown that CGG size-dependent toxicity may be related to elevated mRNA levels of the gene *ASFMR1*, which spans the CGG expansion in the antisense direction. *ASFMR1* has several transcriptional start sites located at 5′ of the CpG island and in intron 2 of *FMR1* [12]. While the role of CpG island methylation 5′ of the expansion in *FMR1* silencing has been characterized, there is minimal understanding of the role of the epi-genotype beyond the CpG island in PM- and FM-related disorders.

Our previous findings suggest that the *FMR1* promoter is much larger than originally thought [3,13,14,15,16], expanding on the 5′ side to an epigenetic boundary located ~800 bp upstream of the CGG expansion [3], which is consistent with the findings of Naumann et al. [17] reporting the location of this boundary to be 65–70 CpG pairs upstream of the CGG repeat expansion. This boundary is located within a region that we named Fragile X-Related Epigenetic Element 1 (FREE1) [3,18], which also contains one of the *ASFMR1* transcription start sites [12,19]. By examining methylation within the *FMR1* ‘minimal’ promoter [20] and the adjacent regions in transformed lymphoblast cell lines from FXS individuals, ‘high-functioning’ individuals with UFM alleles, and controls, we identified new regions differentially methylated in individuals with *FMR1*-related disorders [3,16], with Fragile X-Related Element 2 (FREE2) located largely within *FMR1* intron 1 and Fragile X-Related Element 3 (FREE3) located within *FMR1* intron 2. This study characterized the normal and pathological methylation of these regions in venous blood and brain tissues to: (1) determine the associations between differential methylation within these regions and the presence of FM and PM alleles and X-inactivation and (2) provide novel avenues for the detection of expanded *FMR1* alleles through DNA methylation analyses.

## 2. Results

### 2.1. The FMR1 Methylation Map

In this study, we define the amplicon of the previously described FREE2 region as FREE2(A) [3], with methylation presented in units as described in our earlier publications [15,16,17,18,19]. The amplicons extending 3′ from FREE2(A) were named FREE2(B), FREE2(C), FREE2(D), FREE2(E), and FREE3 (Figure 1). A more detailed genomic organization of the targeted regions, including the number of differentially methylated CpG sites examined, is described in the additional files (Appendix A).

### 2.2. Inter-Group Methylation Comparisons in the Venous Blood of Males

Inter-group comparisons of methylation were performed between 73 males with FXS and FM alleles, 27 males with mosaicism for FM and PM alleles, and 21 males with mosaicism for FM alleles and smaller expansions. Of the CGG size mosaics, 23 were mosaic for the PM and FM alleles, 4 were unusual mosaic cases for either normal-sized FM alleles (29/1300 CGG and 30/175 CGG) or intermediate-sized FM alleles (41/800 CGG and 50/763 CGG). There was a significant increase in intron 1 methylation extending ~1.5 kb 3′ from the CGG expansion to a novel epigenetic boundary in these FM groups, compared with that in 27 control males with normal-sized alleles, 39 males with PM, and 5 with unmethylated FM (UFM) alleles (*p* < 0.0001) (Figure 2 and Appendix A). At the 3′ boundary, three CpG units of FREE2(E) showed a progressive increase in methylation in the control, PM, and UFM groups. However, in all FM groups other than UFM, the difference in methylation between these three CpG units of FREE2(E) was not as large when compared with the control and PM groups. This suggests that there is a transfer of methylation from the adjacent 3′ regions that may be caused by the presence of FM expansions, contributing to loss of the 3′ epigenetic boundary in intron 1 in males with FXS.

This postulate was consistent with the ‘high-functioning’ males with UFM alleles expressing normal or elevated levels of *FMR1* being completely unmethylated within the CpG island [6,7,8] and within intron 1 upstream of the 3′ boundary. Two out of five UFM males showed methylation of intron 1 units (B)CpG2 and (C)CpG4 marginally above the male control range. It is of note that one of the males with a UFM of 323–517 CGGs was the first ‘high-functioning’ male with an FM allele reported to develop FXTAS [8]. He also showed the highest methylation for (B)CpG2 and (C)CpG4 of all individuals with a UFM allele (Figure 2E).

### 2.3. Inter-Group Methylation Comparisons in the Venous Blood of Females

There was a significant increase in intron 1 methylation, extending ~1.5 kb 3′ from the CGG repeat to a novel epigenetic boundary for 132 females with an FM compared with 75 female controls and 157 females with a PM (*p* < 0.0001) (Figure 3 and Appendix A). In contrast to the males with FM alleles, for the females with FM alleles, methylation of this region for most CpG units was not significantly increased in the CGG size mosaic group compared with the female controls and females with a PM (Appendix A). The female control baseline levels of methylation within this intron 1 region were also different from the males, with the female control methylation output ratio (MOR) being between ~0.1 and ~0.5. This is in line with the previously reported levels of methylation resulting from X inactivation [21], also suggesting that X inactivation-sensitive CpG sites extend downstream from the expansion up to the 3′ epigenetic boundary. While methylation also increased above the control range at the 3′ epigenetic boundary for most of females with an FM (Figure 3C), for the females with a PM, methylation at €CpG4, (E)CpG5, a€(E)CpG6, methylation was significantly decreased compared with the controls (*p* < 0.0001). This suggests that methylation in this region is sensitive to X inactivation and presence of PM and FM alleles in females (Appendix A).

### 2.4. Abnormally Decreased Methylation of the Intron 1 3′ Boundary and Intron 2 Sites in Males with a PM Allele

The males with a PM had significantly decreased methylation at the 3′ epigenetic boundary compared with the males with an FM and the male controls (*p* < 0.0001) (Figure 4). This PM-specific decrease was observed for the comparisons of the male controls to (1) all 39 males with a PM collapsed into one group (Figure 4) and (2) the male PM group split into two independent cohorts based on the site of participant recruitment (US cohort *n* = 21; Australian cohort *n* = 18) (Figure 5). Moreover, the (E)CpG4 unit 3′ of this epigenetic boundary showed the greatest overlap in methylation values between the controls and PM ranges in males, with intergroup comparison showing a decrease in methylation of marginal significance in the PM group compared with the male controls (*p* = 0.047).

The intron 2 (F3)CpG1 unit within FREE3 showed an inverse methylation pattern to that observed for most intron 1 sites within FREE2 and to the methylation status of the *FMR1* minimal promoter within the CpG island 5′of the CGG expansion. This inverse pattern involved a significant decrease in methylation in FM only and the groups mosaic for the PM and FM alleles (*p* < 0.0001). The MOR for these ranged between 0.6 and 0.95, compared with the MOR range between 0.9 and 1 for the control, PM, and UFM male groups (Figure 4E). Interestingly, for intron 2 (F3)CpG2, the PM group showed a significant decrease in methylation compared with the male controls (*p* < 0.0001) and males with a UFM (*p* = 0.04). However, there was no significant change in methylation for intron 2 (F3)CpG2 for the PM group compared with the FM group (Figure 4F). Furthermore, (F3)CpG2 methylation was also significantly decreased in the males with a PM and FM mosaic groups compared with the male controls (*p* < 0.0001).

### 2.5. Inter-Group Methylation Comparisons in Brain Tissues and Blood

To determine if the presence of FM alleles is associated with abnormal methylation of the intron 1 and 2 regions in the prefrontal cortex (PFC), we compared the methylation in the brain tissues of males with FXS and male controls with normal-sized alleles. In the PFC, the 3′ epigenetic boundary was lost in the males with FXS, with the methylation output ratio elevated (MOR approaching one) compared with the male controls (MOR < 0.1) (Figure 6A,B). All CpG sites within the FREE2 region within *FMR1* intron 1 5′ of the 3′ boundary were hypermethylated in the PFCs of the males with FXS compared with the controls. The male with FXS with the lowest level of methylation in the PFC across all of the intron 1 sites of all males with only FM alleles was the only mosaic for PM and FM alleles in this group.

To determine if X inactivation is associated with increased methylation within these loci between different areas of the brain and the peripheral tissues, we compared the methylation in this region between male and female control groups in the venous blood, PFC, and cerebellum (Figure 6C,D and Figure 7). The mean methylation output ratio in the venous blood and all brain tissues of the female controls was significantly elevated for 40 CpG sites, snapping across a 1.5 kB region within *FMR1* intron 1 in the female controls compared with the male controls. In contrast, the methylation of two CpG sites—(E)CpG4 and 5 5′of the 3′epigenetic boundary in intron 1 and the (F3)CpG1 site at the 5′ end of *FMR1* intron 2 (FREE3: *ASFMR1* promoter)—was significantly decreased in the PFCs and cerebella of the female controls compared with the corresponding brain tissues in the male controls. This difference, however, was not preserved in the venous blood comparisons (Figure 6C,D).

## 3. Discussion

This study used the EpiTYPER system on the DNA from venous blood and brain tissues to characterize, at the single CpG level, the methylation of *FMR1* introns 1 and 2. The methylation patterns within these intragenic regions were compared between males and females to define the regions sensitive to X inactivation and between individuals with PM and FM alleles and controls to provide novel avenues for the detection of expanded *FMR1* alleles through DNA methylation analysis. These regions are different from the *FMR1* CpG island and the ‘minimal’ *FMR1* promoter described by Kumari and Usdin [20]. Most recently, we have shown that the methylation of CpG sites 6 or 7 and 10–12 at the 5′ end of the *FMR1* intron 1 were most informative in predicting the likelihood of adult females with a PM exhibiting dysexecutive and psychiatric symptoms [11]. This study defined additional regions differentially methylated within *FMR1* introns 1 and 2 due to the presence of expanded *FMR1* alleles in the blood and brain tissues to provide: (1) better understanding of the differentially methylated regions, which may be involved in the regulation of *FMR1* and *ASFMR1* expression, and (2) new targets for the development of biomarkers for DNA methylation-based screening for expanded *FMR1* alleles, which may also have prognostic utility for *FMR1*-associated disorders.

### 3.1. Evidence Suggesting the FMR1 Promoter Is Larger Than the FMR1 CpG Island including FMR1 Intronic Sequences

The *FMR1* promoter, which is sensitive to methylation changes, has been previously suggested to be primarily located within the CpG island upstream of the CGG expansion, as reviewed in [22]. This was based on limited studies showing that *FMR1* regulatory regions which are functionally influenced by methylation changes are located within this region [20,23]. Kumari and Usdin [20] defined the 5′ end of the ‘minimal’ *FMR1* promoter to be 355 bp upstream of the CGG expansion and its 3′ end to be 60 bp downstream of the CGG expansion. This ‘minimal’ promoter was suggested to encompass the *FMR1* CpG island consisting of the 52 CpG sites 5′ of the CGG expansion and 4 *FMR1* transcription start sites [23]. Moreover, Kumari and Usdin [20] suggested that this promoter encompasses the CGG expansion and a portion of *FMR1* exon 1. There was no evidence presented by this group for differential methylation of the *FMR1* intron 1 or 2 sequences in individuals with expanded *FMR1* alleles nor any suggestion of it being included as part of the *FMR1* promoter.

We demonstrated that increased methylation of the *FMR1* intron 1 sites, specifically CpG10–12, located within FREE2(A) was associated with an increased CGG size within the FM range in males, with its analytical superiority to methylation-sensitive Southern blot (used in current FXS diagnostics) and FMRP immunostaining in the blood as a predictor of cognitive impairment in females with an FM [3,14,21] and the dysexecutive-psychiatric phenotype in adult females with PM alleles. Interestingly, a 5′ 380–400 base pair CTCF binding region within intron 1 (*ChIP-seq K562 Sg1* and 2 from ENCODE/University of Washington; 2012) is located in ~100 base pairs 3′ of the intron 1/exon 1 boundary and overlaps with the CpG10–12 at the 5′ end of FREE2 that was clinically most informative in our earlier FM and PM studies (Figure 1). Furthermore, it has been recently shown that as few as one abnormally methylated CpG site strategically positioned at an exon/intron boundary can impact the methylation of the upstream promoter region, gene expression, and the severity of the phenotype at other disease-related loci [24]. Methylation of the FREE2 CpG sites proximal to the intron 1/exon 1 boundary in FM females could interfere with splicing of *FMR1* intron 1 and result in intron retention.

Of potential functional relevance to our observations are also previous reports of abnormal methylation of exonic CpG sites at other loci inhibiting the binding of CTCF, with exon exclusion resulting from aberrant splicing [25]. Evidence showing the formation of RNA:DNA duplexes within this region and that disruption of these duplexes can reactivate the *FMR1* gene in FXS cell lines also suggests that *FMR1* intron 1 upstream of the 3′ boundary should be considered as part of the *FMR1* promoter [26]. Together with our previous findings and those from this study, this evidence indicates that the boundaries of the *FMR1* promoter and the model of epigenetic regulation of *FMR1* should be redefined to include *FMR1* intron 1 CpG sites expanding up to the 3′ epigenetic boundary targeted by the FREE2(E) assay (Figure 2 and Figure 3).

### 3.2. Hypomethylation of FMR1 Intron 2: Novel FXS Biomarkers

Earlier, we found that the CpG sites proximal to the *ASFMR1* transcription start site within *FMR1* intron 2, which we named FREE3 [16], were hypomethylated in FXS lymphoblast cell lines lacking FMRP, with reduced *ASFMR1* expression compared with the controls and UFM cell lines [16]. Here, we showed that in the primary cells from the blood, the same CpG sites also had significantly decreased methylation in the FM, PM/FM CGG size and methylation mosaic males and females compared with the control, PM, and UFM groups. In addition, consistent with our previous and current observations of FXS-specific hypomethylation in intron 2, Alisch et al. [27] described the results for one CpG site within intron 2 that showed FXS-specific hypomethylation, which partially overlapped with the range of methylation values for the controls. In this study, we found that (F3)CpG1 showed the most significant decrease in methylation in the FM FXS groups in both males and females compared with the controls and compared with the ‘high-functioning’ males with UFM alleles (Appendix A). This suggests that decreased methylation of intron 2, specifically for the sites proximal to the *ASFMR1* promoter, are novel epigenetic biomarkers hypomethylated in individuals with FM alleles.

### 3.3. Hypomethylation of the 3′ Epigenetic Boundary in Individuals with a PM

This is the first study to identify several epigenetic biomarkers at the novel 3′ epigenetic boundary that are differentially methylated in all males with a PM. We found that in males with a PM, methylation of the 3′ boundary was decreased compared with the control group, regardless of whether all 39 PM males were considered as one group or split into two independent cohorts based on the site of recruitment. Females with a PM also showed significantly decreased methylation at the 3′ boundary and several CpG sites downstream of the boundary compared with the controls ((E)CpG4, 5, and 6) (Appendix A). These markers therefore have the potential to be used to screen for the presence of PM alleles using DNA methylation testing.

### 3.4. Differentially Methylated Regions Associated with Expanded FMR1 Alleles and X Chromosome Inactivation Conserved between Tissues

This study has demonstrated the following: (1) The 3′ epigenetic boundary identified by the FREE2(E) assay is lost in the brain tissues (as in the venous blood) in males with FXS but not in the controls with normal-sized alleles. All CpG sites within the FREE2 region 5′ of the 3′ epigenetic boundary were hypermethylated in the examined brain tissues of males with FXS compared with the controls, as in the venous blood. (2) The mean methylation for 40 CpG sites spanning across a 1.5 kB region within *FMR1* intron 1 at the 3′ epigenetic boundary in the venous blood and all brain tissues of the female controls with alleles in the normal range was significantly elevated compared with those tissues from the male controls. This strongly suggests that X inactivation causes an increase in DNA methylation across tissues, spanning from FREE2(A) to the 3′ epigenetic boundary identified by the FREE2(E) assay. (3) Differential methylation sensitive to X inactivation within *FMR1* intron 1 up to the 3′ epigenetic boundary was conserved between different regions of the brain in the female controls (prefrontal cortex and cerebellum). This observation highlights the functional importance of the 3′ epigenetic boundary identified in this study.

Interestingly, the CpG sites downstream of the 3′ epigenetic boundary within *FMR1* introns 1 and 2 showed hypomethylation in the females compared to the male controls. This, however, was not preserved in the blood, suggesting that this X inactivation-specific hypomethylation was restricted to the brain tissues. In the male controls, these CpG sites also showed significantly decreased methylation in the brain tissues compared with the venous blood of the male controls. Together, these fundings suggest that the 3′ epigenetic boundary defined in this study is of functional significance, playing a role as an ‘insulator’ to prevent a ‘methylation spillover’ from the gene bogy of *FMR1* to the *FMR1* promoter conserved across different tissues. This postulate is in line with the location of a CTCF binding site immediately at 5′ of this boundary (Figure 1), which has been described to act as such an insulator in other settings [17]. 

### 3.5. Limitations

While this study has several strengths, an important limitation is that no data were available on the level of *FMR1* or *ASFMR1* mRNA, FMRP, or the clinical status (other than for males with a UFM), especially in relation to the presence and severity of *FMR1*-associated disorders in the individuals with PM and FM alleles included in this study. Moreover, functional studies were not performed to examine how loss of the 3′ epigenetic boundary impacts the binding of CTCF and expression of *FMR1*, *ASFMR1*, and FMRP. Future studies should explore (1) the clinical significance of this PM-specific 3′ boundary hypomethylation and how it may be related to the levels of *FMR1* and *ASFMR1* RNA toxicity, as well as the levels of FMRP in different tissues, and (2) the functional significance of its loss and gain in different cell types.

Another limitation is the relatively small number of brain tissues included in this study. Because FXS is a rare disorder, the numbers (4 FXS for PFC and 12 controls for PFC and cerebella) are reasonable to support the main conclusions, with large-sized effects regarding: (1) conservation of the 3′ epigenetic boundary between tissues in the controls examined, (2) its complete loss due to the presence of an FM, and (3) its relationship with X-inactivation. The relatively small sample size for brain tissues included in this study should be also viewed in light of previous studies [28,29] examining molecular changes in brain tissues from individuals with FXS and the controls, which had a comparable number of brain samples used. Future studies should re-examine our observations in larger cohorts and in different brain regions than those examined in this study.

### 3.6. Conclusions

In summary, this study has characterized 45 novel epigenetic biomarker sites where differential methylation is associated with the presence of expanded *FMR1* alleles (but not in high-functioning males with UFM alleles) and X inactivation. These patterns of methylation were also conserved between blood and brain tissues based on comparisons between individuals with FM alleles affected by FXS and neurotypical controls, as well as between the controls of different sexes. Importantly, almost all males with a PM were found to have methylation levels of the 3′ epigenetic boundary in their venous blood below the control range. Significantly decreased methylation of the CpG sites proximal to this boundary was also observed in females with a PM. Therefore, the methylation status of the 3′ boundary and proximal sites may provide a novel way to screen for PM alleles using DNA methylation testing and may have potential diagnostic and prognostic relevance to *FMR1*-associated disorders, which should be examined in future studies.

While FXTAS, FXPOI, and other comorbidities have been linked to PM alleles [30,31,32,33], these are of incomplete penetrance, with no avenues currently available to predict reliably who is most at risk. Unfortunately, this study did not have formal assessment or clinical status information available for individuals with PM alleles that would have allowed examination of the relationships between the phenotype(s) and the levels of hypomethylation of the 3′ epigenetic boundary. Future studies should explore the clinical significance of this PM-specific 3′ boundary hypomethylation and how or if it may be used as a prognostic biomarker to predict who is most at risk of developing PM-associated disorders.

## 4. Materials and Methods

### 4.1. Cohort Description

This study examined methylation in DNA samples from a total of 557 participants, which were collected as parts of previous studies [8,14,21,34]. The cohort consisted of 185 males (aged from birth to 82 years) and 372 females (aged from birth to 80 years). For the males, there were 20 controls (CGG < 40), 39 PM males, 5 UFM males (identified through cascade testing and determined to be unmethylated under Southern blot analysis), 73 FXS-affected FM males (identified through investigation of developmental delay or ASD and 100% methylated through Southern blot analysis), 21 FM methylation mosaics, and 27 PM/FM size mosaics under Southern blot analysis. Formal cognitive assessments were performed for three of the UFM males using the Wechsler Intelligence Test, which is appropriate for chronological age, with a Full-Scale IQ (FSIQ) between 71 and 81, as described previously [8,14,35]. For the females, there were 75 controls (CGG < 40), 157 PM females, 132 FM females (identified through investigation of developmental delay and cascade testing), and 8 were mosaics for PM and FM alleles under Southern blot analysis. For 527 individuals with expanded FMR1 alleles and controls, blood DNA was collected as part of FXS cascade testing and routine molecular microarray testing through VCGS and the Greenwood Genetic Center as described previously [21]. These samples were de-identified before use in this study and as such did not have information on formal assessments or clinical history to allow for accurate assessment of heterogeneity in the clinical picture of the presented groups. An additional 30 females with an FM were recruited through the VCGS. All control brain tissues were provided by the Victorian Brain Bank Network with brain tissues from 4 males affected by FXS from earlier studies [28,29] provided by Flora Tassone from the Department of Biochemistry and Molecular Medicine at the University of California’s Davis School of Medicine in Sacramento, California, USA. Details on the postmortem brain tissue processing, age, and the clinical phenotype are given in Appendix A. This study received ethical approval by The Royal Children’s Hospital Human Research Ethics Committee (single site reference numbers HREC 34227 and HREC 33066 and multi-site HREC reference number HREC/13/RCHM/24, approved on 24 May 2013).

### 4.2. Molecular Analyses

Processing of the venous blood DNA samples and the CGG repeat sizing using PCR was conducted as previously described [36]. For samples with greater than 55 repeats, methylation-sensitive Southern blotting was performed as previously described [3]. Briefly, 7–9 µg of DNA was digested with EcoRI and NruI (located within FMR1 CpG island), and the StB12.3 probe was used to estimate the range of FMR1 allele CGG sizes and CpG island methylation. For the males, Southern blot methylation was used to classify the expanded alleles as either unmethylated, partially methylated, or fully methylated. For the alleles that were estimated to have greater than 150 CGGs and methylated by Southern blotting, the classification of FM was given. The PM allele classification was given to alleles estimated to be between 55 and 200 repeats that were unmethylated by Southern blotting. FMR1 intron 1 and intron 2 methylation was assessed in the same samples using the EpiTYPER system as previously described [3]. The primer sequences and the PCR conditions are listed in Appendix A.

### 4.3. Data Analyses

Testing for normality distribution of MOR was conducted using the Shapiro–Wilk test at a significance level *p* = 0.05. For intergroup comparisons, depending on the results of this test, we used either a two-sample *t* test for the means if the data were normal or a nonparametric Mann–Whitney test for the median if the data were not normal. All analyses were conducted using the publicly available R statistical computing package [37].

## Figures and Tables

**Figure 1 ijms-24-10712-f001:**
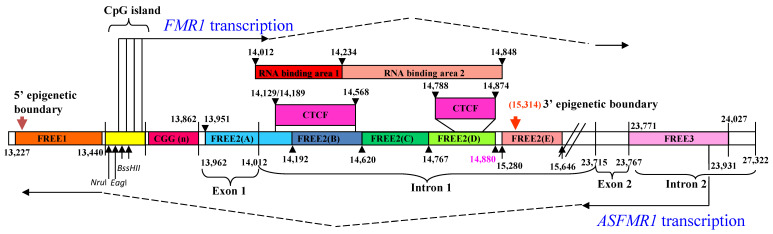
Genetic and epigenetic organization of the 5′ portion of the *FMR1* locus (sequence numbering from GenBank L29074) and locations of target amplicons. The intron and exon regions 5′ and 3′ of the *FMR1* CGG expansion are presented in relation to *FMR1* and *ASFMR1* transcription start sites (the broken lines indicate spliced out regions), Fragile X-Related Epigenetic Elements 1 and 2 (FREE1 and FREE2), the *FMR1* CpG island, and methylation-sensitive restriction sites (*NruI*, *EagI*, and *BssHII*) analyzed using routine fragile X Southern blot testing. A CGG repeat is located within 5′ (UTR) of the *FMR1* gene. *ASFMR1* spans the CGG expansion in the antisense direction and is also regulated by another promoter located in intron 2 of *FMR1*. FREE2 is located downstream of the CGG expansion. The FREE3 region is located within intron 2 of *FMR1* downstream of the second *ASFMR1* promoter. Primers utilized for MALDI-TOF methylation analysis targeted six regions at the Xq27.3 locus, designated as FREE1, FREE2(A) (described as amplicon 5 in Godler et al., [3]), FREE2(B), FREE2(C), FREE2(D), FREE2(E), and FREE3 (color-coded). Red boxes indicate regions pulled down by ChIRP to show RNA binding to intron 1 DNA in the control and FXS human embryonic stem cells (hESCs) at day 45 of differentiation, with RNA area 1 showing the greatest binding intensity (Appendix A). The FREE2 sequences amplified by forward and reverse primers used in the chromatin isolation by the RNA purification (ChiRP) technique are indicated in Appendix A.

**Figure 2 ijms-24-10712-f002:**
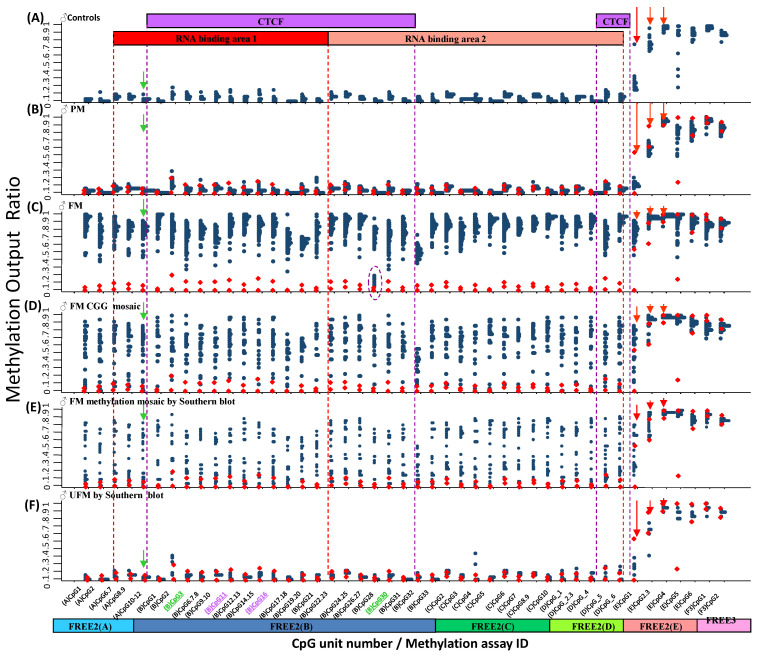
*FMR1* methylation in venous blood 3′ of the CGG expansion in males. (**A**) Male controls with CGG < 40 (*n* = 20). (**B**) Males with a PM (*n* = 39). (**C**) Males with an FM and FXS (100% methylated by Southern blot) (*n* = 73). (**D**) Males with an FM and CGG size mosaicism (*n* = 27). (**E**) Males with an FM and incomplete methylation through Southern blot (*n* = 21). (**F**) ‘High-functioning’ males (*n* = 5) with a UFM identified by Southern blot. Red dots overlayed onto plots (**B**–**F**) represent the upper and lower normal methylation range (two standard deviations from the mean methylation of male controls with normal CGG size alleles). On the x axis, the (B)CpG3 and (B)CpG30 units (in green) had fragments of the same mass. (B)CpG11 and (B)CpG16 (in purple) also had fragments of the same mass. The EpiTYPER mass spectrometry approach could only provide the mean methylation for these fragments of the same mass. The green arrow indicates methylation of CpG units 10–12, previously described to be significantly associated with the type and severity of cognitive impairment in a female carrier of expanded *FMR1* alleles. The purple oval indicates methylation of a CpG unit which separates the male FM group into two distinct groups. The red arrows indicate methylation CpG units at the novel 3′ epigenetic boundary. The purple boxes represent the CTCF binding sites. The red boxes indicate regions pulled down by ChiRP to show RNA binding to DNA in the control and FXS hESCs in earlier studies (Appendix A).

**Figure 3 ijms-24-10712-f003:**
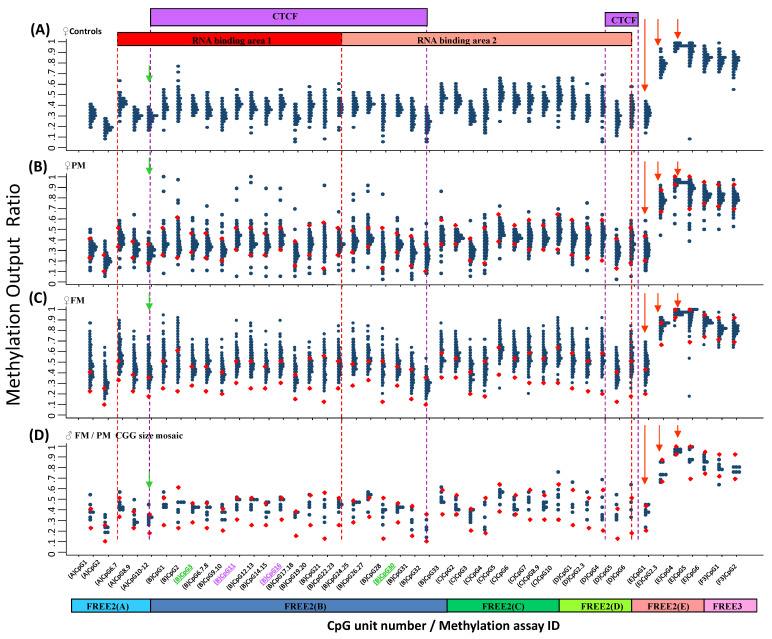
*FMR1* methylation in venous blood at 3′ of the CGG expansion in females. (**A**) Female controls with CGG < 40 (*n* = 75). (**B**) Females with a PM (*n* = 157). (**C**) Females with an FM (*n* = 132). (**D**) Female mosaic for PM and FM alleles (*n* = 8). The red dots overlayed onto plots (**B**–**D**) represent the upper and lower normal methylation range (two standard deviations from the mean methylation of the female control group with normal size CGG alleles). On the x axis, the (B)CpG3 and (B)CpG30 units (in green) had fragments of the same mass. (B)CpG11 and (B)CpG16 (in purple) also had fragments of the same mass. The EpiTYPER mass spectrometry approach could only provide mean methylation for these fragments of the same mass. The green arrow indicates methylation of CpG units 10–12, which we previously described to be significantly associated with the type and severity of cognitive impairment in a female carrier of expanded *FMR1* alleles. The purple oval indicates methylation of a CpG unit, which separates the female with an FM into two distinct subgroups. The red arrows point to methylation of CpG units at the novel 3′ epigenetic boundary. The purple boxes represent CTCF binding sites. The red boxes indicate regions pulled down by ChiRP to show RNA binding to DNA in the control and FXS hESCs in earlier studies (Appendix A).

**Figure 4 ijms-24-10712-f004:**
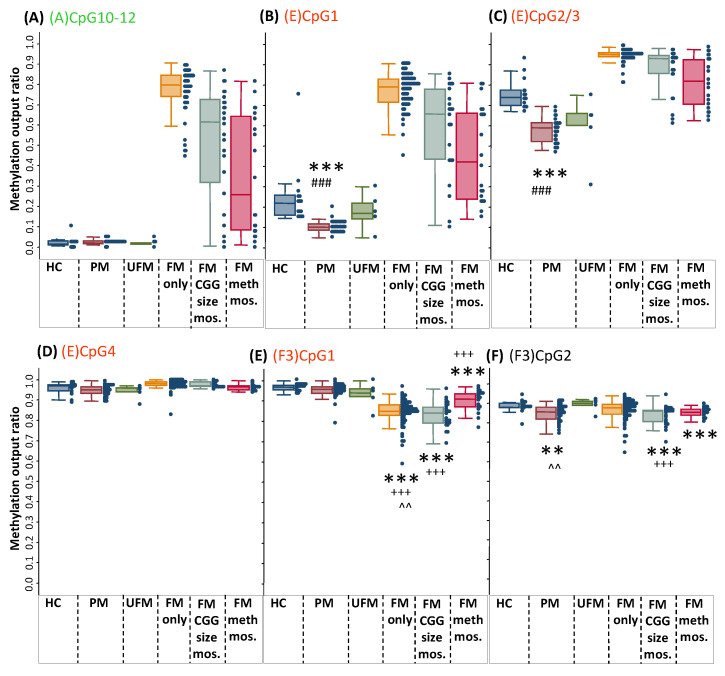
*FMR1* methylation at the exon1/intron 1 boundary, the intron 1 3′ epigenetic boundary, and the intron 2 region overlapping with the *ASFMR1* promoter in venous blood of males. Nonparametric tests for intergroup comparisons were performed in males for methylation of 6 CpG units for 21 control males with CGG size < 40 (HC), 39 males with a PM, 5 ‘high-functioning’ males with a UFM, 73 males with an FM affected by FXS (FM only; 100% methylated by Southern blot), 27 males with an FM CGG size mosaicism (including PM/FM, GZ/FM, and normal size/FM), and 21 males with an FM partially methylated from Southern blot assessments (FM meth. mos.) (*n* = 21). (**A**) FREE2 CpG10–12. (**B**–**D**) FREE2 3′ epigenetic boundary CpG 1–4 sites. (**E**,**F**) FREE3 CpG1 and 2. Note: Selected comparisons of significant decrease in methylation *p* < 0.0001 are indicated by *** compared with HC, ### compared with FM, and +++ compared with PM. For ^^, *p* < 0.05 compared with UFM, and ** is *p* < 0.05 compared with HC. The exact p values for these intergroup comparisons are presented in Appendix A.

**Figure 5 ijms-24-10712-f005:**
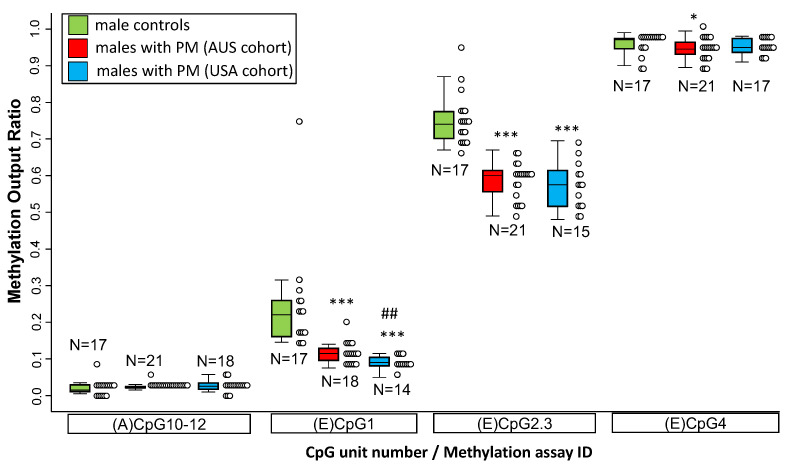
PM-specific hypomethylation of the 3′ epigenetic boundary within *FMR1* intron 1 in venous blood of males. Comparison of methylation at FREE2 (A)CpG10–12 for the exon1/intron 1 border and the 3′ epigenetic boundary within intron 1 between control males (HC) and males with a PM from the USA cohort (CGG55–130) and the Australian cohort (CGG57–170). Note: *** *p* < 0.0001; * *p* < 0.05 compared with HC; ## *p* < 0.05 compared with the USA PM cohort. The exact *p* values for these intergroup comparisons are presented in Appendix A.

**Figure 6 ijms-24-10712-f006:**
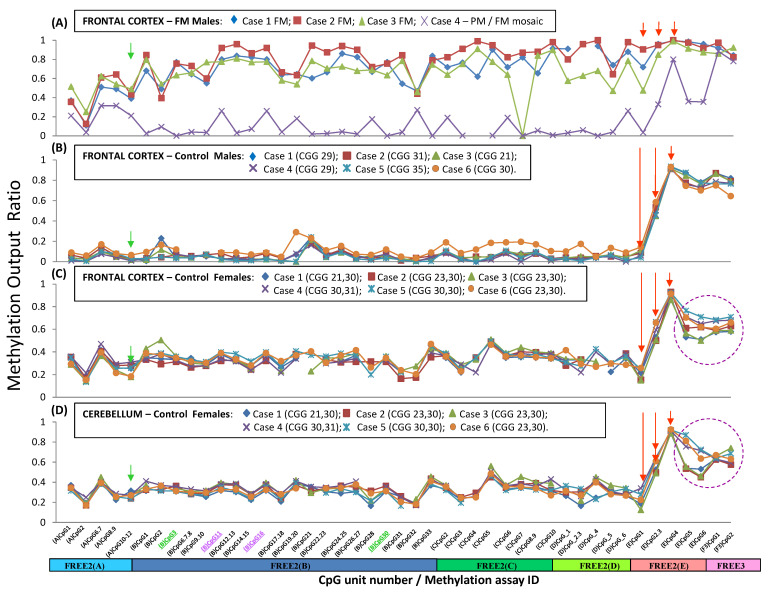
*FMR1* methylation in brain tissues 3′ of the CGG expansion in males and females. (**A**) Prefrontal cortexes of males with an FM (*n* = 4). (**B**) Prefrontal cortexes of male controls (CGG < 40) (*n* = 6). (**C**) Prefrontal cortexes of female controls (CGG < 40) (*n* = 6). (**D**) Cerebella of female controls (CGG < 40) (*n* = 6). On the x axis, the (B)CpG3 and (B)CpG30 units (in green) had fragments of the same mass. (B)CpG11 and (B)CpG16 (in purple) also had fragments of the same mass. The EpiTYPER mass spectrometry approach could only provide the mean methylation for these fragments of the same mass. The green arrow indicates methylation of CpG units 10–12, which we previously described to be significantly associated with the type and severity of cognitive impairment in a female carrier of expanded *FMR1* alleles. The red arrows point to methylation of CpG units at the novel 3′ epigenetic boundary.

**Figure 7 ijms-24-10712-f007:**
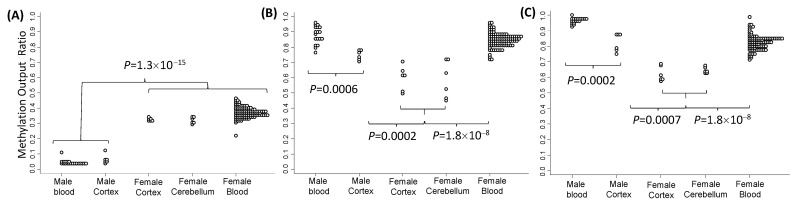
*FMR1* intron 1 and 2 methylation pattern variation in postmortem brain tissue and blood in individuals with *FMR1* CGG allele sizes in the normal range. Mean methylation output ratio in prefrontal cortex (PFC) from six male controls, PFCs and cerebellum were from six female controls and venous blood of 75 female and 20 male controls across (**A**) 40 CpG sites snapping across a 1.5 kB region within *FMR1* intron 1, (**B**) 2 CpG sites (E)CpG4 and 5 5′ of the intron 1 epigenetic boundary, and (**C**) (F3)CpG1 at the 5′ end of *FMR1* intron 2 (*ASFMR1* promoter).Interestingly, methylation of two CpG sites (E)CpG4 and five at 5′ of the 3′ epigenetic boundary in intron 1, as well as the (F3)CpG1 site at the 5′ end of *FMR1* intron 2 (*ASFMR1* promoter), was significantly decreased in the PFCs of the male controls compared with the venous blood of the male controls (Figure 6C,D). This suggested presence of PFC-specific methylation for these CpG sites.

## Data Availability

The data that support the findings of this study are available on request from the corresponding author. The data are not publicly available due to privacy or ethical restrictions.

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
