# Peer review of "Defining the 3′Epigenetic Boundary of the FMR1 Promoter and Its Loss in Individuals with Fragile X Syndrome"

_ijms, 2023, doi:10.3390/ijms241310712_

Round 1

Reviewer 1 Report

This study is largely self referenced as also proven by the large number of self citations. Specific points:

1. It is not clear how many  cytosines that may undergo methylation are present in each so-called FREE region. This should be specified.

2. One ASFMR1 promoter is lvcated in intron 2, not in exon 2, as indicated in Fig. 1 (which is difficult to read).

3. Conclusions drawn from brain analyses are unwarranted, given the small number of samples (4 FXS brains vs. 12 controls).

4. The 5' methylation boundary (ref. 21) is not at 1 kb from the CGG sequence as stated by the authors, but at 2.5 kb.

5. In the material and methods section it is stated that Southern blot analysis gives an estimate of the number of CGGs. It should be stated that Southern blot can indicate  a range

Quality of English generally good. Some improvement in the clarity of the prose is possible.

Author Response

Dear Professor Pop,

Thank you for the opportunity to submit a revised manuscript, titled:

Defining the 3’epigenetic boundary of the FMR1 promoter and its loss

in individuals with fragile X syndrome”

We would like to thank the editor and the reviewers for their constructive feedback.

All changes to the revised manuscript are outlined in detail in our response to the reviewers, with changes included as track changes within the body of the manuscript. The page numbers referred to in the response relate to the track changed version of the manuscript.

Yours sincerely,

David Godler, PhD

Associate Professor/Group Leader, Diagnosis and Development,

Murdoch Children’s Research Institute

50 Flemington Road

Parkville, VIC Australia 3052

Ph: +61 3 8341 6496

E: [email protected]

Reviewer #1 (Comments to the Author):

Comment 1: This study is largely self referenced as also proven by the large number of self citations.

Author Response: This study characterizes FM, PM and X-inactivation specific methylation changes within the Fragile X Related Epigenetic Element (FREE) regions in blood and brain tissues. As such, the paper includes previous literature examining FREE regions in other cohorts and settings including Colak et al. 2014 Science, Alisch et al. 2013 BMC Med Genet and Naumann et al. 2009 Am J Hum Genet. If the reviewer is aware of other ‘non-self’ referenced literature examining FREE regions, we would be open to including these into the manuscript. We were the first to define FREE regions in 2009 and have performed considerable research focusing on these regions since that time. As such, it is unavoidable not to include these references in this manuscript, as there are no other alternative references we are aware of to demonstrate the points made.

Comment 2: It is not clear how many cytosines that may undergo methylation are present in each so-called FREE region. This should be specified.

Author Response: These details are provided in Note S1, with additional comment now included in the results section P3, lines 111 and 112, in response to this reviewer comment referring to this information. Please also see Discussion section P17, Lines 88 to 90: ‘Mean methylation for 40 CpG sites spanning across a 1.5kB region within FMR1 intron 1 the 3’ epigenetic boundary in venous blood and all brain tissues of female controls with alleles in the normal range was significantly elevated as compared these tissues from to male controls.’

Comment 3: One ASFMR1 promoter is located in intron 2, not in exon 2, as indicated in Fig. 1 (which is difficult to read).

Author Response: The figure 1 legend has been modified to address this comment and be in line with the figure 1 indicating that the ASFMR1 promoter is located in intron 2. The size of the figure has also been increased to improve readability.

Comment 4: Conclusions drawn from brain analyses are unwarranted, given the small number of samples (4 FXS brains vs. 12 controls).

Author Response: This comment has been now addressed in the Discussion section, P17, Lines 117 to 125.

Comment 5: The 5' methylation boundary (ref. 21) is not at 1 kb from the CGG sequence as stated by the authors, but at 2.5 kb.

Author Response: The 5’ methylation boundary where there is no distinct difference between methylation in individuals with FM alleles and controls is located ~800 bp upstream of the CGG expansion (see Godler et al 2010 Hum Mol Genet. Fig 2, CpG1 to 10 of Amp 4; and Fig 3). This is consistent with the location of the boundary reported by Naumann et al ‘65 to 70 CpG pairs upstream of the CGG repeat in the human FMR1 gene’ (Naumann et al Discussion P 611). Propriate changes have been made on P2, lines 90-91.

Comment 6: In the material and methods section it is stated that Southern blot analysis gives an estimate of the number of CGGs. It should be stated that Southern blot can indicate  a range

Author Response: This has now been addressed in the Methods section, P18, Lines 166 to 168, indicating that Southern blot was used to estimate the range of FMR1 allele CGG sizes.

Reviewer 2 Report

Please see the attached review.

Author Response

Dear Professor Pop,

Thank you for the opportunity to submit a revised manuscript, titled:

Defining the 3’epigenetic boundary of the FMR1 promoter and its loss

in individuals with fragile X syndrome”

We would like to thank the editor and the reviewers for their constructive feedback.

All changes to the revised manuscript are outlined in detail in our response to the reviewers, with changes included as track changes within the body of the manuscript. The page numbers referred to in the response relate to the track changed version of the manuscript.

Yours sincerely,

David Godler, PhD

Associate Professor/Group Leader, Diagnosis and Development,

Murdoch Children’s Research Institute

50 Flemington Road

Parkville, VIC Australia 3052

Ph: +61 3 8341 6496

E: [email protected]

Reviewer #2 (Comments to the Author):

Comment 1: I would recommend adding information about the clinical picture in the presented groups with different genetic/epigenetic results.

Author Response: This information has been included on P19, Lines 159 to 168: ‘Formal cognitive assessments were performed for three of the UFM males using the Wechsler intelligence test appropriate for chronological age, with Full Scale IQ – FSIQ between 71 and 81, as described previously [11, 18, 33]. For females there were 75 controls (CGG<40), 157 PM, 132 FM (identified through investigation of developmental delay and cascade testing), and 8 were mosaic for PM and FM alleles by Southern blot analysis. For 527 individuals with expanded FMR1 alleles and controls, blood DNA was collected as part of FXS cascade testing and routine molecular microarray testing through VCGS and the Greenwood Genetic Center as described previously [28]. These samples were de-identified before use in this study and as such did not have information on formal assessments or clinical history to allow for accurate assessment of heterogeneity in the clinical picture of the presented groups.’

For males with UFM alleles where formal assessment data were available, we have described relationships with the clinical picture. Please P3, Lines 129 to 135,

Please also see P18, Lines 143 to 150.

 Comment 2: Moreover, I would suggest adding a paragraph with a summary of possible novel implications for the diagnostic process, and whether they may be used in a routine practice.

Author Response: This has been included on P18, lines 143 to 135.
